# Effects of Yachting Motivation on Yachting Satisfaction and Re-Yachting Intention

**DOI:** 10.3390/bs14040272

**Published:** 2024-03-25

**Authors:** Jaehyun Ha, Dongwook Cho

**Affiliations:** College of Physical Education, Keimyung University, Daegu 42601, Republic of Korea; jaehyunha@kmu.ac.kr

**Keywords:** yachting motivation, yachting satisfaction, re-yachting intention, yachting tourism

## Abstract

It is clear that yachting tourists are motivated to spend their leisure time at sea. However, few studies have determined the relationship between tourists’ motivation and satisfaction with yachting, and re-yachting intention. Furthermore, the purpose of this study was to examine the relationships among yachting motivation, yachting satisfaction, and re-yachting intention. More specifically, this study sought to examine whether yachting motivation influenced yachting satisfaction, and determine if re-yachting intention was affected by yachting motivation. In order to analyze the data for this study, a total of 240 questionnaires were distributed to yachting tourists at three private yacht club operators, utilizing the convenient sampling method. A total of 219 usable questionnaires were analyzed for this study, while 21 questionnaires were discarded due to incompletion of the survey. Data were analyzed with multiple regression analysis using SPSS version 27.0. The results revealed that entertainment, stress reduction, and family/friends were positively and significantly related to yachting satisfaction. However, socializing and external ostentation did not contribute to yachting satisfaction. Secondly, the results indicated that entertainment, socializing, and stress reduction had a positive and significant effect on re-yachting intention. Conversely, external ostentation and family/friends had no significant influence on re-yachting intention. It is necessary for yacht club business operators to recognize the yachting motivation of the participants. It is also recommended that appropriate marketing strategies are implemented to increase yachting tourism, which would possibly influence yachting satisfaction and re-yachting intention.

## 1. Introduction

With the remarkable increase in leisure time in modern society, leisure activities can be considered more important than work. It is a fact that more people are enjoying leisure activities such as recreation, exercise, sport participation, and tourism [1]. Among the various leisure activities, yachting is one of the popular ways to spend leisure time at sea by sailing or on a motor yacht [2]. Thus, yachting tourism constitutes a type of special interest for participants that contributes significantly to the marine economy [2,3]. The International Council of Marine Industries Association [4] indicated that there were approximately 15.7 million yachts (33% of the global total) in the U. S., and around 5 million yachts in Europe, mainly in Germany, Poland, Sweden, Netherlands, and Finland. As such, yachting tourism has dominated in North America and Europe. In recent years, developed Asia–Pacific countries have also become attractive yachting tourism destinations owing to increased attention [5].

To better understand yachting tourists’ decision to cruise on a yacht, it is essential to recognize the factors that contribute to yachting tourism. Previous research in recent decades has been conducted regarding the factors that affect yachting tourism [3,6,7,8]. Research has suggested that yachting motivation, yachting satisfaction, and re-yachting intention might have a significant influence on yachting tourism, implying that they might be the most reliable indicators of actual yachting [9,10,11].

Despite the fair amount of research that has been devoted to yacht tourism, little attention has been paid to the psychological factor that motivates yachting tourists, and its effect on yachting satisfaction and re-yachting intention. According to Funk, Alexandris, and McDonald [12], it was believed that a psychological factor was one of the significant influences on a participant’s decision. The participants’ psychological factor was labeled as motivation. Motivation was “an internal force that directs behavior toward the fulfillment of needs” [13]. More specifically, internal motivation refers to engaging in an activity with the desire to participate that comes from within the individual, driven by factors such as interest, enjoyment, or a sense of personal satisfaction. Extrinsic motivation, on the other hand, involves engaging in an activity in order to earn external rewards or avoid punishment. The motivation comes from external factors rather than inherent enjoyment of the activity itself [13]. Wann and his associates [14] have suggested that motivation was one of the important elements with respect to the decision-making processes and actual behavior. The effect of motivation on satisfaction and behavioral intention has been emphasized in previous research. Ahn and Shin [9] proposed and examined a model regarding tourists’ satisfaction with yacht activities. They found that intrinsic motivation had a strong influence on satisfaction judgments and formations. Bosque and Martin [15] and Iso-Ahola [16] examined the tourist satisfaction model and suggested that intrinsic motivation acted as an antecedent to satisfaction.

In addition to the likely effect of motivation on satisfaction, motivation would be predictive of behavioral intention. More specifically, Prayag and Grivel [17] have suggested that a tourist’s motivation to attend a festival event might function as the determinant of behavioral intention. While there are a number of studies related to the motivation of tourists and its impact on satisfaction and future consumption behavior, minimal empirical research has been conducted on yachting motivation and its impact on yachting satisfaction and re-yachting intention.

Even though many determinants have been used as indicators of yachting satisfaction and re-yachting intention, this study was undertaken to focus specifically on yachting motivation as the most relevant and predictable variable of yachting satisfaction and re-yachting intention. Therefore, the purpose of this study was to examine the influence of yachting motivation on yachting satisfaction and re-yachting intention. It is believed that this study can contribute to advancing the knowledge of yachting tourist consumption behavior, providing a more comprehensive understanding of the relationship between yachting motivation, and yachting satisfaction, and re-yachting intention. 

## 2. Materials and Methods

### 2.1. Participants

The sample was collected from yachting tourists at three private yacht clubs in the southeastern region of South Korea. A convenience sampling method was used to collect the data for this study. The proper approval was obtained from the Institutional Review Board (IRB) of a higher education institution. All participants agreed to the recognition of anonymity and voluntary participation by signing the informed consent form. A total of 240 questionnaires were distributed and returned from the three different places in South Korea. Among them, 21 questionnaires were excluded because they were not reliable and applicable. There were 219 usable questionnaires. Table 1 represents the demographic characteristics of the sample. The sample comprised 73.1% males (n = 160) and 26.9% females (n = 59) with a mean age of 44.5 years, ranging from 21 to 60. The sample indicated that 79.5% were married (n = 174) and 20.5% were unmarried (n = 45). The participants’ educational backgrounds were broken down as follows: 23.3% had a high school degree (n = 51), 27.4% a college degree (n = 60), 42.5% an undergraduate degree (n = 93), and 6.8% a graduate degree (n = 15). For monthly income (KRW), 53.9% (n = 116) earned over 4,000,000. Regarding participants’ frequency of yachting in a year, one time (48.4%, n = 106), two times (21.9%, n = 48), three times (14.2%, n = 31), and more than four times (15.5%, n = 34) were represented, respectively.

### 2.2. Instruments

The survey instrument for this study was slightly modified to address the inadequacies of the existing scales, which did not fit the context of this study. Modifications were deemed necessary. A panel of experts, including two professors and two doctors in the marine sport tourism program, was asked to examine the questionnaires thoroughly to determine the content validity of the survey instrument. Based on the information the panel provided, the survey instrument was revised.

Yachting motivation consisted of five sub-dimensions, including entertainment, stress reduction, family/friends, socializing, and external ostentation. The items of yachting motivation were modified from the work of James and Ross [18], and Yao, Lie, and Huang [3]. Each sub-dimension of yachting motivation consisted of three items. Three items of yachting satisfaction were adapted from the previous studies of Laverie and Arnett [19], Yoon, Lee, and Lee [20], and Yoon and Uysal [21]. Three items of re-yachting intention were modified from the previous studies of Chalip, Green, and Hill [22], Prayag and Grivel [17], and Ha, Park, and Han [23] (Table 2). All items except the demographic profiles (gender, marital status, age, education, monthly income, and frequency of yachting in a year) were measured using a seven-point Likert scale ranging from 1 (strongly disagree) to 7 (strongly agree).

### 2.3. Data Analysis

Data analysis was performed using SPSS version 27.0. The data were analyzed with frequency analysis, descriptive analysis, correlation analysis, reliability analysis, exploratory factor analysis (EFA), and multiple regression analysis. First, frequency analysis was used to determine the demographic profiles of the sample. Second, means and standard deviations of all the variables were calculated with the use of descriptive statistics. Third, the correlation values between all the variables were estimated with the use of Pearson correlation coefficient analysis to identify if multicollinearity existed between all the variables. Fourth, reliability was analyzed through the estimation of Cronbach’s alpha to determine the internal consistency of the items for each variable. Fifth, exploratory factor analysis was employed to identify a set of dimensions and measures for subsequent analysis. Lastly, multiple regression analysis was conducted to identify the relationships among yachting motivation, yachting satisfaction, and re-yachting intention. 

## 3. Results

### 3.1. Correlation Values and Descriptive Statistics

The correlation values among all the variables are examined in Table 3, which displays the correlation values, means, and standard deviations of each variable. All the correlation values ranging from 0.680 to 0.192 were significantly correlated yet distinct, indicating that significant levels of multicollinearity did not exist because the correlation values among all the variables were less than 0.85 [24].

### 3.2. Exploratory Factor Analysis

Principal component analysis with direct oblimin rotation for exploratory factor analysis was performed on 15 items that were proposed to measure yachting motivation. The magnitude of the Kaiser–Meyer–Olkin (KMO) measure of sampling adequacy for the fifteen items was 0.871, indicating that the input correlation matrix data were adequate for exploratory factor analysis. The coefficient of the Bartlett test of sphericity was statistically significant (χ^2^ = 3561.259, *p* < 0.001). The results of the two examinations indicated the data were appropriate for exploratory factor analysis [25]. As reported in Table 4, the fifteen individuals’ items were loaded on five separate factors with Eigenvalues larger than 1.0, which collectively accounted for 90.256% of the total variance.

The fifteen items were retained as five distinct factors providing a measure of yachting motivation. The following factors were identified: External ostentation (three items), Socializing (three items), Stress reduction (three items), Entertainment (three items), Family/Friends (three items). Factor loading ranged from 0.916 to 0.794. More specifically, the factor loading of each factor was 0.915 to 0.882 for external ostentation, 0.916 to 0.890 for socializing, 0.881 to 0.857 for stress reduction, 0.884 to 0.838 for entertainment, and 0.863 to 0.794 for family/friends. The reliability of each factor was measured with Cronbach’s alpha. Cronbach’s α coefficients for all the factors were larger than the cut-off value of 0.70, ranging from 0.959 to 0.921 [26]. The results of the Cronbach’s α coefficients revealed that the items of each factor were internally consistent.

Exploratory factor analysis was performed on the six individual items proposed to measure yachting satisfaction and re-yachting intention. The KMO for the six items was 0.824. The coefficient of the Bartlett test of sphericity was statistically significant (χ^2^ = 1843.733, *p* < 0.001). As reported in Table 5, the six individuals’ items were loaded on two separate factors with Eigenvalues larger than 1.0, which collectively accounted for 94.594% of the total variance. 

The six items were retained as two distinct factors. Three items of yachting satisfaction and three items of re-yachting intention were identified. Factor loading of yachting satisfaction ranged from 0.948 to 0.941. Factor loading of re-yachting intention ranged from 0.942 to 0.923. The reliability of yachting satisfaction (0.979) and re-yachting intention (0.963) was larger than the cut-off value of 0.70. 

### 3.3. Relationship between Yachting Motivation and Yachting Satisfaction

Multiple regression analysis was performed to examine the relationship between yachting motivation and yachting satisfaction. Table 6 represents a summary of the results for five dimensions of yachting motivation predicting yachting satisfaction. The regression model is significant (F = 29.159, *p* < 0.001, R^2^ = 0.406), indicating that a significant amount of variance in yachting satisfaction is explained by the five dimensions of yachting motivation. More specifically, entertainment (β = 0.223, *p* < 0.01), stress reduction (β = 0.330, *p* < 0.001), and family/friends (β = 0.168, *p* < 0.05) had a significantly positive impact on yachting satisfaction. However, socializing and external ostentation did not contribute to yachting satisfaction.

### 3.4. Relationship between Yachting Motivation and Re-Yachting Intention

Table 7 represents a summary of the results for the five dimensions of yachting motivation predicting re-yachting intention. The regression model is significant (F = 66.934, *p* < 0.001, R^2^ = 0.611), indicating that a significant amount of variance in re-yachting intention is explained by the five dimensions of yachting motivation. More specifically, entertainment (β = 0.369, *p* < 0.001), stress reduction (β = 0.287, *p* < 0.001), and socializing (β = 0.153, *p* < 0.01) had a positive and significant impact on re-yachting intention. However, family/friends and external ostentation had no significant influence on re-yachting intention.

## 4. Discussion

The results of this study indicated that entertainment and stress reduction had a positive impact on yachting satisfaction and re-yachting intention. That is, entertainment and stress reduction were likely to function as determinants of yachting satisfaction and re-yachting intention. These findings suggest that entertainment and stress reduction were more salient to and associated with yachting satisfaction and re-yachting intention than other yachting motivations. Highly satisfied yachting tourists value enjoyable experiences and an escape from personal, social, and/or psychological pressures. It has been suggested that these two motivations represent a core set of psychological needs that explains the yachting tourists’ desire to participate in yachting activities.

More specifically, the results of the values for standardized regression coefficients revealed that entertainment was the strongest predictor of yachting satisfaction. This finding was consistent with Mullin, Hardy, and Sutton [27] in that those enjoyable experiences served to bolster satisfaction of participation. It seemed reasonable that entertainment explained the most variance in yachting satisfaction because the response to such entertaining experiences were usually expressed in the form of yachting satisfaction. The results of the values also revealed that stress reduction was the strongest predictor of re-yachting intention. The findings confirmed the study of Ahn and Shin [9], which revealed that stress reduction was the most influential discriminator of behavioral intention regarding yachting activities. 

With regard to the relationship between family/friends and yachting satisfaction, this study found that family/friend togetherness was positively associated with yachting satisfaction. This result was consistent with previous studies [12,14]. That is, as an individual was motivated to participate in yachting activities because it provided an opportunity to spend time with family members and/or friends, he/she would be more likely to be satisfied with yachting activities.

Regarding socializing, there was no relationship between socializing and yachting satisfaction. This result was not supported in the previous studies by Ahn and Shin [9], and Prayag and Grivel [17], where the socializing motive was a strong predictor of satisfaction with active sport tourism. Additionally, numerous studies outside of the sport tourism context have indicated that socializing was one of the participation motivations that significantly influenced participant satisfaction. However, because of a lack of empirical examinations regarding socializing as a component of yachting motivation and its influence on yachting satisfaction, caution was required when interpreting the effect of socializing on yachting satisfaction. Thus, further research is deemed necessary to identify the empirical standpoint. 

With regard to the relationship between socializing and re-yachting intention, it was found that socializing was a salient feature of re-yachting intention. This finding was consistent with Funk, Alexandris, and McDonald [12], which revealed that the levels of intention to attend sport events varied according to the expectation of group affiliation, indicating that the expectation to meet new people and spend time with them was one of the most influential discriminators of revisit intention. For yachting tourists, it was precisely the social nature of yachting that had attracted them. They were motivated by the socializing motive, that is, a desire to meet new people and spend time with others.

The results of this study reveal that the family/friends motive fails to explain a meaningful variance in re-yachting intention. According to Yao, Zheng, and Parmak [28], family/friends’ motivation was identified as an important factor determining behavioral intention regarding yachting tourism. James and Ross [18] identified family/friend ties as an important factor determining sport consumption behavior. The result of this study is not consistent with these previous studies. Thus, to better understand the relationship between family/friend motivation and re-yachting intention, it is proposed that further research is necessary.

Regarding external ostentation, the results of this study reveal that external ostentation does not directly contribute to yachting satisfaction and re-yachting intention. These results were dissimilar to the results of Park and Yang [29], indicating that external ostentation had a significantly discriminant function that distinguished levels of satisfaction and behavioral intention. While external ostentation failed to explain meaningful amounts of variance in yachting satisfaction and re-yachting intention, it was not entirely meaningless. That is, although it did not directly influence yachting satisfaction and re-yachting intention, they still shared variance with each other and all of them interacted, as shown by the correlation matrix.

As such, it is crucial for yacht club business operators to understand the yachting motivations and develop proper marketing strategies considering their effect on yachting satisfaction and re-yachting intention. Increasing yachting satisfaction and re-yachting intention is important for yacht club business operators because yachting satisfaction and re-yachting intention strongly impact the actual behavior of yachting tourists. The results of this study confirmed that yachting satisfaction and re-yachting intention were influenced by two yachting motives (entertainment and stress reduction). Based on these findings, a number of practical implications emerged from this study, namely that in order to increase the level of yachting satisfaction and re-yachting intention, yacht club business operators need to focus primarily on creating, fostering, and maintaining the entertainment and stress reduction levels of yachting tourists. Therefore, promotional efforts should be designed to strengthen yachting tourists’ feelings of pleasure and excitement, and to represent a desire for individuals’ mental well-being through opportunities to escape and remove themselves from the daily work and routines that create tension and stress.

This study suggests several limitations and directions for future research. First, since this study utilized yachting tourists from three private yacht clubs using a convenient sampling method, the results of this study might be limited in their generalizability. Thus, it is recommended that future research should be conducted with the use of a random sampling method. Second, according to the suggestions from previous studies, situational and environmental factors might either complement or contradict an individual’s satisfaction and behavior [12,27]. It is suggested that future research should include the aforementioned variables in the model and examine the relationship among them.

## 5. Conclusions

This study proposed that yachting satisfaction and re-yachting intention could be predicted by yachting motivation, which was composed of five sub-dimensions: entertainment, stress reduction, family/friends, socializing, and external ostentation. The results of this study indicated that yachting satisfaction was positively and directly influenced by entertainment, stress reduction, and family/friends. However, socializing and external ostentation had no influence on yachting satisfaction. The results of this study also revealed that there were positive and direct effects of entertainment, stress reduction, and socializing on re-yachting intention. However, family/friends and external ostentation did not contribute to re-yachting intention.

The findings from this study should be considered by practitioners or professionals in the development of strategies to attract more yachting tourists based on the results of yachting motivation, yachting satisfaction, and re-yachting intention. Moreover, practitioners or professionals should recognize the different aspects of participation in yachting tourism. Based on the findings, it can be helpful to provide marketing events or programs in which participants can easily get together and have more opportunities for social interaction with family and friends. Also, it might be beneficial for yachting practitioners to plan a program to reduce participants’ stress and enhance their entertainment.

## Figures and Tables

**Table 1 behavsci-14-00272-t001:** The results of demographic characteristics.

Demographic Information	Classification	Frequency (n = 219)	Percent (%)
Sex	Male	160	73.1
	Female	59	26.9
Marital Status	Married	174	79.5
	Unmarried	45	20.5
Age	20–29	9	4.1
	30–39	37	16.9
	40–49	119	54.3
	50–59	52	23.7
	Over 60	2	0.9
Education	High school degree	51	23.3
	College degree	60	27.4
	Undergraduate degree	93	42.5
	Graduate degree	15	6.8
Monthly Income (KRW)	Under 2,000,000	16	7.3
	2,000,000–2,900,000	40	18.2
	3,000,000–3,900,000	45	20.5
	4,000,000–4,900,000	54	25.6
	Over 5,000,000	62	28.3
Frequency of Yachting in a Year	1	106	48.4
	2	48	21.9
	3	31	14.2
	Over 4	34	15.5

**Table 2 behavsci-14-00272-t002:** Questionnaire subscales and items.

Subscales	Items
External ostentation	1. Yachting enables me to bring the fame.
	2. Yachting is a good opportunity to let me know to others.
	3. By yachting skills, I get other people’s attention.
Socializing	1. Yachting enables me to have social interaction with other.
	2. Yachting provides a great opportunity to make friends.
	3. Yachting helps me to develop a close relationship with others.
Stress reduction	1. Yachting relieves daily stress.
	2. Yachting feels like an escape from my hectic daily life.
	3. Yachting contributes to my emotional well-being.
Entertainment	1. I get excited when I ride on a yacht.
	2. I feel the passion when I was yachting.
	3. I do not realize how time passes when I’m on a yacht.
Family/Friends	1. Yachting is a great way to spend time with friends.
	2. Yachting is a great way to spend time with family.
	3. I feel a close relationship with friends/family when playing yacht.
Yachting satisfaction	1. I am satisfied with playing yacht.
	2. I am satisfied with my decision to play yacht among the various leisure activities.
	3. Yachting gives me a sense of accomplishment.
Re-yachting intention	1. I will continue to play yacht in the future.
	2. I will actively consider yachting when planning my leisure activities.
	3. Playing yacht next time is something I plan to do.

**Table 3 behavsci-14-00272-t003:** The results of correlation values and descriptive statistics.

	1	2	3	4	5	6	7
2	0.510 **	1					
3	0.549 **	0.583 **	1				
4	0.441 **	0.353 **	0.404 **	1			
5	0.450 **	0.522 **	0.490 **	0.192 **	1		
6	0.509 **	0.566 **	0.509 **	0.322 **	0.386 **	1	
7	0.680 **	0.637 **	0.575 **	0.474 **	0.494 **	0.495 **	1
M	4.681	4.866	5.001	5.287	4.237	5.179	4.698
SD	1.205	1.174	1.321	1.143	1.368	1.193	1.293

** *p* < 0.01. Note: 1 = Entertainment; 2 = Stress reduction; 3 = Family/Friends; 4 = Socializing; 5 = External ostentation; 6 = Yachting satisfaction; 7 = Re-yachting intention.

**Table 4 behavsci-14-00272-t004:** The results of the exploratory factor analysis and reliability of each factor of yachting motivation.

Items	Factor Loading				
External ostentation 1	**0.915**	0.047	0.185	0.141	0.189
External ostentation 2	**0.894**	0.069	0.210	0.194	0.153
External ostentation 3	**0.882**	0.042	0.244	0.130	0.204
Socializing 1	0.066	**0.916**	0.112	0.167	0.154
Socializing 2	0.028	**0.908**	0.101	0.183	0.110
Socializing 3	0.057	**0.890**	0.160	0.192	0.163
Stress reduction 1	0.226	0.149	**0.881**	0.207	0.222
Stress reduction 2	0.238	0.173	**0.866**	0.185	0.250
Stress reduction 3	0.251	0.118	**0.857**	0.187	0.240
Entertainment 1	0.160	0.153	0.127	**0.884**	0.216
Entertainment 2	0.198	0.246	0.229	**0.846**	0.250
Entertainment 3	0.164	0.281	0.247	**0.838**	0.229
Family/Friends 1	0.200	0.151	0.245	0.215	**0.863**
Family/Friends 2	0.220	0.189	0.220	0.202	**0.818**
Family/Friends 3	0.194	0.171	0.258	0.291	**0.794**
Reliability	0.950	0.938	0.959	0.950	0.921
Eigenvalue	7.675	2.285	1.276	1.260	1.042
Variance	51.164	15.236	8.503	8.403	6.949

**Table 5 behavsci-14-00272-t005:** The results of exploratory factor analysis and reliability of yachting satisfaction and re-yachting intention.

Items	Factor Loading	
Yachting satisfaction 1	**0.948**	0.253
Yachting satisfaction 1	**0.945**	0.275
Yachting satisfaction 1	**0.941**	0.253
Re-yachting intention 1	0.226	**0.942**
Re-yachting intention 2	0.283	**0.925**
Re-yachting intention 3	0.263	**0.923**
Reliability	0.979	0.963
Eigenvalue	4.292	1.383
Variance	71.537	23.057

**Table 6 behavsci-14-00272-t006:** The results of multiple regression analysis for yachting satisfaction.

Factor	β	SE	β	t	*p*
Constant	1.473	0.353		4.169	0.000
Entertainment	0.221	0.069	0.223 **	3.210	0.002
Stress reduction	0.335	0.072	0.330 ***	4.638	0.000
Family/Friends	0.152	0.065	0.168 *	2.322	0.021
Socializing	0.037	0.064	0.035	0.578	0.564
External ostentation	0.022	0.057	0.025	0.379	0.705

R = 0.637, R^2^ = 0.406, Adjusted R^2^ = 0.392. * *p* < 0.05, ** *p* < 0.01, *** *p* < 0.001.

**Table 7 behavsci-14-00272-t007:** The results of multiple regression analysis for re-yachting intention.

Factor	β	SE	β	t	*p*
Constant	−0.475	0.310		−1.532	0.127
Entertainment	0.396	0.060	0.369	6.574	0.000 ***
Stress reduction	0.316	0.063	0.287	4.989	0.000 ***
Family/Friends	0.091	0.057	0.093	1.587	0.114
Socializing	0.173	0.056	0.153	3.101	0.002 **
External ostentation	0.098	0.050	0.104	1.951	0.052

R = 0.782, R^2^ = 0.611, Adjusted R^2^ = 0.602. ** *p* < 0.01, *** *p* < 0.001.

## Data Availability

The datasets generated for this study are available upon request to the corresponding author.

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
