# Peer review of "Effects of Yachting Motivation on Yachting Satisfaction and Re-Yachting Intention"

_behavsci, 2024, doi:10.3390/bs14040272_

Round 1

Reviewer 1 Report

Comments and Suggestions for Authors

A very interesting paper, with a clear structure and a well done empirical research. 

However, some aspects need to be clarified:

- is there "one" model the paper is based on, it should be clearly stated, why authors choose that one and did not use others, which might also be used in sports tourism

- regarding motivation cultural aspects play an important role. Where did the participants come from (one country, one region, from all over the world)? - are there differences regarding this aspect 

- same questions comes along with number of Yachting activities per year 

- at least within the discussion these points should be considered regarding the quality of results 

- finally: why did You choose South Korea ? - could the place have influence to the results and if yes, what could be concluded ?

Comments on the Quality of English Language

minor revision is needed

Author Response

Review 1

 A very interesting paper, with a clear structure and a well done empirical research. 

However, some aspects need to be clarified:

- is there "one" model the paper is based on, it should be clearly stated, why authors choose that one and did not use others, which might also be used in sports tourism

Answer) It was widely utilized in the academic field of sports tourism in South Korea which is easier to compare with previous research so that we were able to strengthen our finding.

- regarding motivation cultural aspects play an important role. Where did the participants come from (one country, one region, from all over the world)? - are there differences regarding this aspect 

- same questions comes along with number of Yachting activities per year 

Answer) It might be revealed the different results based on cultural aspects or frequency to play Yachting activities per year. Actually, we are conducting this survey in U.S so that we will be able to compare the cultural aspect and frequency to play Yachting activities per year. However, it’s hard to jump to conclusion as this moment.

- at least within the discussion these points should be considered regarding the quality of results 

Answer) We mentioned it on the limitations and directions for the future research paragraph on the discussion parts.

- finally: why did You choose South Korea ? - could the place have influence to the results and if yes, what could be concluded ?

Answer) As you might know, this study was designed to conduct in South Korea because Korea is a peninsula, we thought it would be relatively easy to recruit research participants compared to other regions.

As mentioned earlier, it might have different results if we conducted the studies in other countries. We mentioned it on the limitations and directions for the future research on the discussion parts.

Dear Reviewer

We are sincerely appreciated your valuable comments and feedback. We did our best to respond your comments but it might not enough to understand your intentions. If so, please let us know that we will re-respond your comments as soon as possible. Again, thank you so much for your time and efforts.

Reviewer 2 Report

Comments and Suggestions for Authors

this paper presents a good research work which however requires some improvement, especially in terms of conceptual support.

In this regard, I suggest delving deeper into the topic of motivation by explaining the difference between intrinsic motivation and extrinsic motivation and explaining better how these types of motivation can influence the perception of satisfaction and behavioral orientations.

this in-depth analysis also allows us to improve the conclusions. they currently only reflect some results already described previously. Instead their task is to provide suggestions by making the results communicate better with the reference theory. In this case, the differences that emerge between intrinsic and extrinsic motivation could make the reader better understand the results obtained from the research.

Furthermore, they could help the authors provide guidance to professionals in the sector and scholars on how to continue these studies.

Author Response

Review 2

this paper presents a good research work which however requires some improvement, especially in terms of conceptual support.

In this regard, I suggest delving deeper into the topic of motivation by explaining the difference between intrinsic motivation and extrinsic motivation and explaining better how these types of motivation can influence the perception of satisfaction and behavioral orientations.

Answer) We provided the difference between intrinsic motivation and extrinsic motivation with some edits on original paper. 

this in-depth analysis also allows us to improve the conclusions. they currently only reflect some results already described previously. Instead their task is to provide suggestions by making the results communicate better with the reference theory. In this case, the differences that emerge between intrinsic and extrinsic motivation could make the reader better understand the results obtained from the research.

Furthermore, they could help the authors provide guidance to professionals in the sector and scholars on how to continue these studies.

Answer) We provided some suggestions on conclusions.

Dear Reviewer

We are sincerely appreciated your valuable comments and feedback. We did our best to respond your comments but it might not enough to understand your intentions. If so, please let us know that we will re-respond your comments as soon as possible. Again, thank you so much for your time and efforts.